# Use of Fire-Extinguishing Balls for a Conceptual System of Drone-Assisted Wildfire Fighting

**Burchan Aydin [1,\*], Emre Selvi [2], Jian Tao [3] and Michael J. Starek [4]**

[1] Department of Engineering & Technology, Texas A&M University-Commerce, 2600 W Neal St., Commerce, TX 75428, USA

[2] Department of Engineering, Jacksonville University, 2800 University Blvd N, Jacksonville, FL 32211, USA; eselvi@ju.edu

[3] Texas A&M Engineering Experiment Station, High Performance Research Computing, and Texas A&M Institute of Data Science, Texas A&M University, College Station, TX 77843, USA; jtao@tamu.edu

[4] School of Engineering and Computing Sciences, Conrad Blucher Institute for Surveying and Science, Texas A&M University-Corpus Christi, 6300 Ocean Drive Unit 5868, Corpus Christi, TX 78412, USA; michael.starek@tamucc.edu

\* Correspondence: burchan.aydin@tamuc.edu; Tel.: +1-903-886-5174

**Abstract:** This paper examines the potential use of fire extinguishing balls as part of a proposed system, where drone and remote-sensing technologies are utilized cooperatively as a supplement to traditional firefighting methods. The proposed system consists of (1) scouting unmanned aircraft system (UAS) to detect spot fires and monitor the risk of wildfire approaching a building, fence, and/or firefighting crew via remote sensing, (2) communication UAS to establish and extend the communication channel between scouting UAS and fire-fighting UAS, and (3) a fire-fighting UAS autonomously traveling to the waypoints to drop fire extinguishing balls (environmental friendly, heat activated suppressants). This concept is under development through a transdisciplinary multi-institutional project. The scope of this paper encloses general illustration of this design, and the experiments conducted so far to evaluate fire extinguishing balls. The results of the experiments show that smaller size fire extinguishing balls available in the global marketplace attached to drones might not be effective in aiding in building fires (unless there are open windows in the buildings already). On the contrary, results show that even the smaller size fire extinguishing balls might be effective in extinguishing short grass fires (around 0.5 kg size ball extinguished a circle of 1-meter of short grass). This finding guided the authors towards wildfire fighting rather than building fires. The paper also demonstrates building of heavy payload drones (around 15 kg payload), and the progress of development of an apparatus carrying fire-extinguishing balls attachable to drones.

**Keywords:** drones; unmanned aircraft system (UAS); fire extinguishing balls; remote sensing; wildfires

## 1. Introduction

The roles of forests in nature can be listed as cleansing water, stabilizing the soil, cycling nutrients, controlling the climate, absorbing carbon dioxide, and producing oxygen. They are the habitats for the wildlife, and an important segment of the country's economic wealth [1]. However, every year millions of acres of forest are lost because of forest fires. The forest fires can be divided into two broad classes; wildfires and prescribed fires. Wildfires are either caused by accidental or malicious acts of human (almost 90% of wildfires in U.S [2]) or by nature (lightning, etc.) (around 10% of wildfires in USA [2]). They are not planned by forest managers and do not occur under controlled settings [2], and pose severe hazards to wildlife and society. According to National

Interagency Fire Center, from 1 January to 30 November 2017 there were 56,186 wildfires in U.S. causing around 9.2 million acres of forest loss [3]. According to Verisk's 2017 wildfire risk analysis, wildfires caused $5.1 billion loss to U.S. within last 10 years [4]. Additionally, 4.5 million U.S. homes are identified at high or extreme risk of wildfire [4]. It is crucial to detect and suppress the wildfires as early as possible; due to the rapid convection spread and long combustion cycle [1]. However, early intervention is not usually possible due to terrain that is hard to access and the impact of wind, intensified by various fuel sources contained in the forests. The fuels of wildfires can be listed as understory foliage, small or large branches, upper layers of the forest floor, and treetop residues [2]. The fuels can also be categorized as surface fuels versus aerial fuels. Surface fuels include combustible material lying on or immediately above the ground, roots and organic soils. Duff, litter, and low-lying vegetative growth are accounted as the principal surface fuels. Duff is composed of layers of partially decomposed organic matter on the forest floor, and has little impact on the forward rate of spread of a fire, but support a slow, smoldering type of combustion [2]. Litter consists of fallen leaves, needles, twigs, bark, cones, and small branches that have not decayed sufficiently. The low-lying vegetation includes grasses, low shrubs, ferns, seedlings, and other small plants. On the other hand, aerial fuels include all live and dead material not in direct contact with the ground such as tree branches and foliage due to volatile oils and resins [2]. Besides the possible fuels existing on forests, it is important to understand the wildfire behavior. There are four descriptors of wildfire behaviors: rate of spread (chains/hour), heat per unit area (Btu/ft$^2$), flame length (feet), and fireline intensity (Btu/ft/s) [5]. Rate of spread is the forward rate of spread at the head of a surface fire, whereas heat per unit area is a measure of heat released by a square foot of fuel within the flaming zone. Flame length is the length from the midway of active flaming zone to the average position of the flame tip, while fireline intensity is the amount of heat released per second by a foot wide slice of the flaming combustion zone. The latter two descriptors are the basis for the fire suppression interpretation [5]. Table 1 shows the suppression interpretations based on the flame length and fireline intensity [5,6]. Heat per unit area is equal to the fireline intensity divided by the rate of spread. Thus, for a given fireline intensity, the faster the rate of spread, the less heat will be directed to the site. On the other hand, a slow moving fire will concentrate substantial heat on the site [7].

**Table 1.** Suppression interpretations

| Flame Length (feet) | Fireline Intensity (Btu/ft/s) | Interpretation |
|---|---|---|
| <4 | <100 | • Fire can be attacked at the head or flanks by persons using handtools.<br>• Handline should hold the fire. |
| 4–8 | 100–500 | • Fires are too intense for direct attack on the head by persons using handtools.<br>• Handline cannot be relied on to hold fire.<br>• Equipment such as plows, dozers, pumpers, and retardant aircraft can be effective. |
| 8–11 | 500–1000 | • Fires may present serious control problems-torching out, crowning, and spotting.<br>• Control effort at the fire head will be ineffective. |
| >11 | >1000 | • Crowning, spotting, and major fire runs are probable.<br>• Control efforts at the head of fire are ineffective. |

The surface fire behavior chart in Figure 1 demonstrates these descriptors. This chart is developed by utilizing the programs "BehavePlus Version 5.0.5 and Fire Characteristics Chart v2.0". Four vegetation types were considered for this surface model as a replication of the model provided by Andrews, et al. (2011) [6]. Legend provided on the figure presents the vegetation models. These models represent different regions of the heat per unit area versus rate of spread graph. Therefore, short grass, timber litter, short needle litter, and chapparal vegetations are included in the experimental design of the conceptual model. The main objective of our research is proposing a

system to assist wildfire fighting mainly for these four vegetation models, due to the fact that they represent all different regions on the rate of spread versus heat per unit area chart as shown in Figure 1. For instance, short grass fires, which are coded as number '1' in the chart, resemble the wildfire behavior of high rate of spread (chains/hr) and low heat per unit area (Btu/ft²). This way, the system could be feasible to be applied for a wide spectrum of vegetation and landscapes.

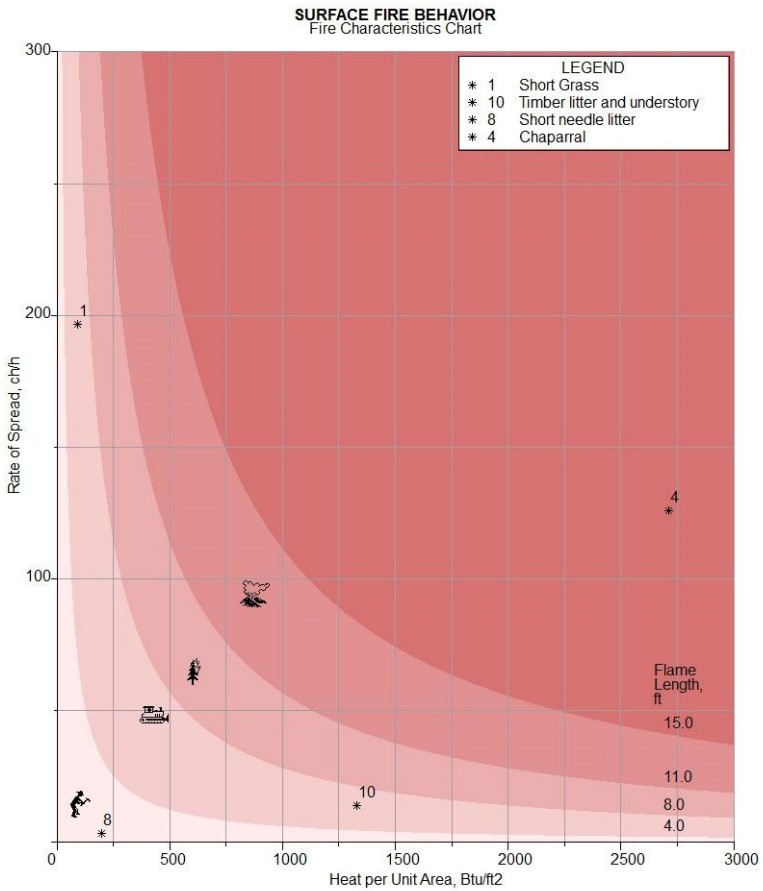

**Figure 1.** Surface fire behavior.

The conceptual design is based on fire extinguishing balls being released from a small unmanned aircraft system (UAS) to mitigate the risks associated with wildfires to manned crafts, firefighter crew, and society. The proposed remote-sensing technology concept can be used to monitor, detect, diagnose, prognose all categories in Table 1, whereas the fire-extinguishing balls can be effective for the categories up to 8 feet. It could essentially be applicable in wildland urban interface zone fires rather than large scale forest fires. A fire extinguishing ball is a sphere-shaped product made of Styrofoam filled with environmentally friendly non-toxic chemical powders. The manufacturers claim that the ball self-activates within at least 3 seconds of contact with the fire; explodes and releases the extinguishing agents [8,9]. There are currently two main brands of fire extinguishing balls in the marketplace: the Elide and the AFO. Elide is made in Thailand, whereas AFO is made in China. The Elide ball is 1.5 ± 0.2 kg, with a diameter of 147 mm [8,9]. AFO balls are available in three different sizes; the smallest ball weighs around 0.5 kg, the medium ball weighs around 0.7 kg, and the largest ball weights around 1.3 kg. The Elide ball is the patented pioneer version with global certifications including ISO 9001:2008. The manufacturers of both brands claim that these balls are effective against fires involving solid burning materials (Class A), flammable liquids and gases (Class B), and energized electrical equipment (Class C) [8,9]. However, no information with respect to effectiveness for wildfires are provided by the manufacturers. Thus, the objective of this initial part of the research was to experiment with the use of these balls for building fires and wildfires. Figure 2 shows the Elide and AFO fire extinguishing balls.

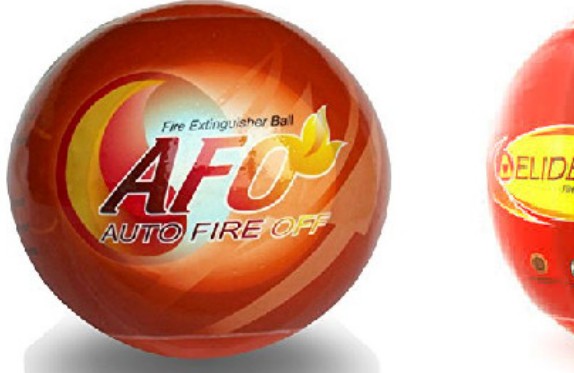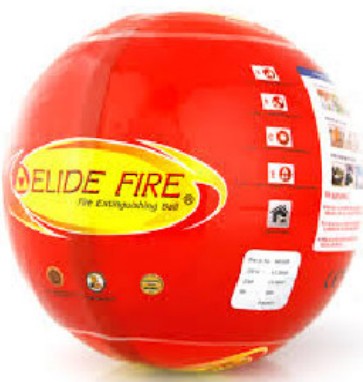

**Figure 2.** AFO and Elide fire extinguishing balls [8,9].

## 2. Background

There is a growing need to use drones with diverse capabilities and various civilian and military applications including search and rescue missions, environmental protection, mailing and delivery, active weapon engagement, space and marine drones, etc. [10]. This paper specifically emphasizes the use of drones in firefighting applications. The time to suppress a forest fire is critical with regards to the fire burden consisting of economic, environmental, and social losses [1]. To decrease the fire burden, currently UASs are used by several fire departments nationwide for search and rescue operations, and for situational awareness assessed by monitoring (finding a potential fire or hot spot), detection (triggering an alarm to inform related operators and personnel), diagnosis (determining the fire's location and extent and tracking its progress), and prognosis (predicting the future of the fire) [11] thanks to the remote-sensing capabilities via incorporated sensors and processing units.

Scientific research on the use of drones have been conducted from both technical and non-technical perspectives in the last decade. The non-technical aspect is based mainly on understanding society's attitudes towards drones. Several survey studies addressed this issue [12–16]. The common finding from these studies is that public perception of drones is dependent on the application. Firefighting applications of drones are well supported by the public. Another non-technical aspect is the economic feasibility and justification. For instance, Laszlo et al., studied the economic effectiveness of using drones at forest fires by proposing three approaches [17]. On the other hand, the technical aspect of using drones for firefighting has focused on remote sensing, simulation studies, and use of various fire suppressants.

UAS-based remote sensing with computer vision is a rapid, mobile, and low-cost alternative or supplement to the traditional sensing technologies; ground-based systems, manned air vehicles, and satellite-based systems. The images retrieved from satellites are not sufficient to create effective firefighting missions due to low temporal and low spatial resolutions, and ineffectiveness in populated areas [18,19]. Ground measurement equipment may suffer from restricted surveillance ranges, and manned aircraft are typically expensive [1]. The firefighters' need for frequent and high-quality information with regards to fire behavior [20] could be achieved by autonomous UAS even under low light and high smoke conditions by means of the embedded sensors with low cost. The evolution of using UAS for fire imaging started with using infrared sensors (cameras) detecting the radiation emitted by fires [21]. Infrared sensors require a direct view of the radiation source for effective results [19]. Visual cameras, on the other hand, have been used to detect the smoke produced by fire under daylight conditions. Visual cameras can mainly provide flame height and angle, and fire's location and width. Fire detection with visual cameras is mainly based on contrast, texture, and motion analysis [19]. Another early method is the use of light detection and ranging (LIDAR) devices to identify the concentration of fire smoke particles [22].

There are several simulation studies showing promising results for swarms of UAS to monitor, detect, diagnose, and prognose wildfires. For instance, Merino et al., presented a system of heterogeneous UAS incorporated with infrared and visual cameras to reduce the number of false fire

alarms by means of data fusion techniques [23,24]. Howden and Hendtlass described an algorithm that performs localized, rather than centralized, control of multiple UASs that surveys complex areas for fires [25]. Considering the results of the simulation runs, their algorithm is proven to be robust for adapting to loss or addition of one or more UASs. Pham et al. (2017) also developed a decentralized algorithm for a team of UASs to track the fire spreading boundaries [26]. Their algorithm ensures that collisions are avoided between UASs. Yuan et al. (2017) developed a forest fire detection model using color and motion-based analysis [11]. The reliability and accuracy of their model in forest fire detection was proven by two experiments; one in a real forest fire video recorded by an aircraft, and one in a real-time indoor fire video gathered by a UAS. Similarly, Alexis et al., (2009) presented a decentralized autonomous system of multiple UASs to monitor the perimeter of the forest fire [27]. Casbeer et al., in a simulation study, developed path planning algorithms utilizing infrared images to track the perimeter of simulated forest fires for both single UAS and multiple UASs scenarios. Multiple UAS algorithms allowed more frequent and detailed updates to the firefighters with respect to the perimeter and location [1]. On the other hand, Lee at al., evaluated five deep convolutional neural networks for detection of wildfires [18]. The majority of the deep convolutional networks were identified to be effective in detecting wildfires by analyzing aerial images. Dios et al., (2011) tested UAS under several scenarios measuring sources of errors in detecting the forest fires with infrared and visual cameras. Their real-time experiments showed promising results for use of UAS incorporating data from several cameras by statistical data fusion methods [19].

Although the ongoing research on using UAS to detect wildfires has shown promising progress, the development of such systems, including software, hardware, and applications, is still at a minimum [1]. Moreover, what we are focusing that is different from the previous researchers' work is developing a remote-sensing capability that will give an alarm not only for spot fires, but also for the detection of the fire head and flanks' risk of spreading to a building, fences, or firefighting crew. Thus, the remote sensing will be designed to detect spot fires, buildings or fences, firefighters, fire behavior, and based on that will recognize the risk of the fire spreading to any building, fences, or firefighter and trigger an alarm with precise location parameters. The design of this remote-sensing system is an ongoing project, but it is out of the scope of this particular paper.

Besides the use of UAS for remote sensing, commercial uses for fire suppression has not been possible due to the restricted payload and flight time capacities and the existing research on using UAS to suppress fire is very limited.

The first set of studies used or recommended the use of water as the fire suppressant. One of these studies was conducted by Lockheed Martin by demonstrating a collaborative system including a UAS and a helicopter. The hot spots identified by the UAS were attacked by the helicopter by dropping water [28]. Several other companies, such as Aerones, Nitrofirex, Singular Aircraft have also been working on developing drone systems that utilize water to suppress building or wild fires [29–31]. Phan and Liu (2008) proposed a system including an airship, UAS and unmanned ground vehicles (UGVs). The airship, which is the top level of the hierarchy, generates a mission plan by utilizing wildfire, UAS, and UGV dynamic models. It sends commands to the UASs and UGVs to travel to certain waypoints for them to suppress the fire. This system has not been tested in real life, but simulation tests have been planned albeit not executed yet [32]. Qin et al. (2017) built a UAS that can autonomously handle the tasks of collecting water, transporting it to the specific fire point, and releasing the water onto the fire. The system was tested in outdoor experiments in both fair and windy conditions successfully [33]. Yet, considering large scale use of these two concepts in the future, with a theme of dropping water, leaves many doubts on how UASs might find water sources to collect water and respond to a fire in a prompt fashion. Even though their systems work well in theory, we believe they would not be practical unless the fire is in a remote area right next to a large water source. Also, the potential water damage to buildings is not negligible.

There are few studies that considered fire suppressants other than water. In a simulation study, Kumar et al. developed a control model for multiple UASs to detect the fire front and also to fight the fire with limitless fire suppressant fluid [34]. They tested their model with use of 10 UASs via

simulation. Yet, their system has not been tested in a real life scenario. Also, the assumption of limitless suppressant fluid is not practical. As undergraduate design projects at Florida International University (FIU), a student team designed a quadcopter with a ball-throwing mechanism that can shoot a single unit of fire-extinguishing ball by using compressed springs [35], whereas another team designed a claw mechanism to drop a unit of fire extinguishing ball [36]. None of the teams tested the mechanism attached to UAS. In another study conducted at FIU, a UAS design and a leader-follower algorithm were proposed to model the use of a swarm of UAS simultaneously dropping fire extinguishing balls to the wildfire [37]. On the other hand, researchers from the University of Nebraska-Lincoln collaborating with the U.S. Geological Survey propose a system where UASs drop ping pong ball-sized "dragon eggs" of fire. Their goal is to destroy the possible fuels before the wildfire approaches [38]. Thus, they are actually starting fire with the fire balls to control wildfires. This way the fuel load is decreased. In case of an unintentional wildfire; the fire intensity will remain low, which results in reduced fire spread, more time for intervention, and less fire burden [39].

Our approach in this research project is similar to the former three studies that recommended throwing fire extinguishing balls into fire, rather than the latter approach of starting controlled fires. Specifically, the purpose of our study is using three emerging technologies concurrently; the UASs, remote sensing, and the so-called fire extinguishing balls to combat wildfires. The proposed system consists of: (1) scouting UAS to detect spot fires and monitor the risk of wildfire approaching a building, fences or firefighting crew via remote sensing, (2) communication UAS to establish and extend the communication channel between scouting UAS and fire-fighting UAS, and (3) a fire-fighting UAS autonomously traveling to the waypoints to drop fire extinguishing balls (environmental friendly, heat activated suppressants). As aforementioned, development of this proposed system is an ongoing multi-institutional, transdisciplinary research project. This paper illustrates the initial part of the research; controlled experiments to examine the effectiveness and efficiency of fire extinguishing balls.

## 3. Materials and Methods

With the aim of determining the fire extinguishing balls' actual effectiveness and efficiency for building fires and/or wildfires, controlled experiments were conducted in collaboration with the local fire department in Commerce, TX. In the first part of the experiments, Class A and Class B fires were ignited by the fire fighters inside an 8 × 8 feet fire demonstration cell as shown in Figure 3a,b respectively. Class C fire was omitted as a suggestion of the fire department; due to the fact that Class C fires turn into Class A fires, instantly. The smallest size of AFO, which is approximately 0.5 kg, was used due to budget limitations. The efficiency was measured by 'time to activate' response variable. The times were recorded by two observers to take averages to reduce data collection errors. A total of four trials for Class A fire and four trials for Class B fire were conducted. Due to the time and labor constraints of the fire department, only two replications could be performed for experimental conditions. In half of the trials, the AFO was thrown directly into fire, whereas they were located in a wall mount for the rest of trials. Table 2 shows the experimental design.

**Table 2.** Experimental design used for the fire-extinguishing balls experiment.

| Fire Classification | Application of the Ball |
|---|---|
| Class A | Wall Mount |
| Class B | Direct Throw |
| Class A | Wall Mount |
| Class B | Wall Mount |
| Class B | Wall Mount |
| Class A | Direct Throw |
| Class A | Direct Throw |
| Class B | Direct Throw |

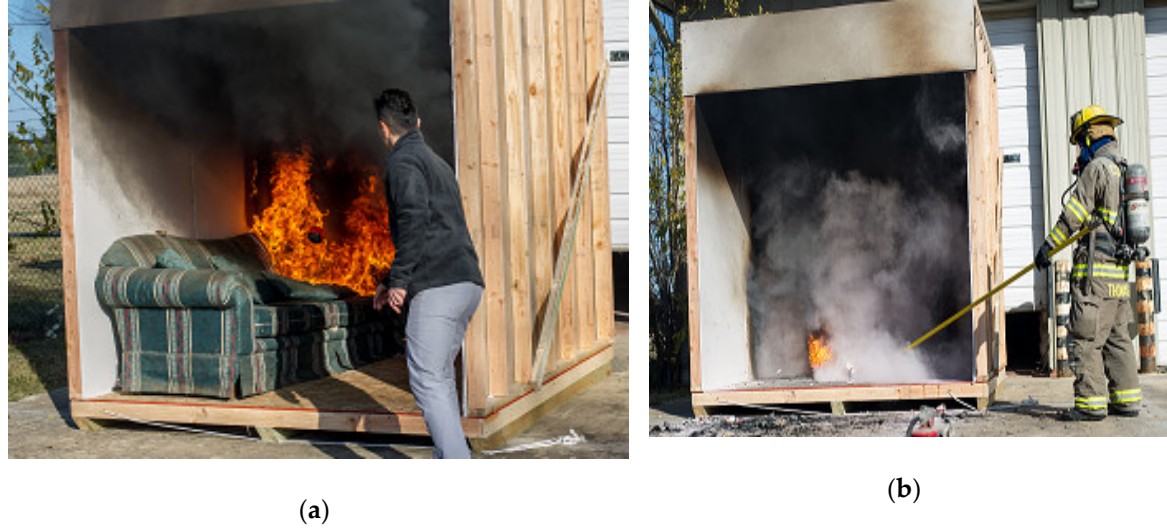

|  |  |
|:---:|:---:|
| (**a**) | (**b**) |

**Figure 3.** (**a**) 'Class A' fire in the demonstration cell for the case study; (**b**) 'Class B' fire in the demonstration cell for the controlled experiments.

The second part of the experiments focused upon wildfires; specifically for short grass fires coded as "1" in Figure 1. As aforementioned short grass fires cause a very high rate of spread (chain/hr) with low heat per unit area (Btu/ft²). The three other main vegetation types presented in Figure 1 (timber litter, short needle litter, and chapparal) were not included in these experiments, but will be encompassed in further studies to complete the testing of the conceptual design. Under direct observance of the local fire department, short grass fire was started for two experimental trials. Once the grass fire reached around one meter diameter, 0.5 kg AFO ball was pushed into it by firefighters. See Figure 4 for the short grassfire experiment. Both trials were conducted at the same time of the day in the afternoon to ensure equal temperature and humidity conditions between trials. If the fire extinguisher ball exploded but was not able to extinguish the fire, or if it did not explode and the short grass fire passed an approximate area of two meters in diameter, the fire was going to be intervened by traditional methods by the firefighters. These conditions were set by the fire department to ensure safety of the experiments.

The results of these experiments are reported in Section 4.

## 4. Results

The AFO ball was efficient in terms of time to activate for both fire classifications, but was not efficient when kept in a wall mount. It took an overall average of 290.9 seconds to explode when on the wall mount. Moreover, no matter if it is directly thrown or kept on a wall mount, it was ineffective with respect to actually extinguishing the building fire. It extinguishes the fire as soon as it explodes, but in a few seconds the fire restarts. Therefore, the firefighters had to extinguish the fire with traditional means in order to avoid complete destruction of the demonstration cell in each experimental run. Further experiments are planned to investigate the Elide fire extinguishing ball under same experimental settings.

The following table (Table 3) shows the data collected in each trial. As mentioned, the time data are the average recordings of two observers.

Contrary to the building fire experiments, the research team got promising results for the wildfire case. It took around same time for the AFO model to explode, and the short grass fire was completely extinguished in both trials. Figure 4 shows the short grass fire before and after explosions.

**Table 3.** Experimental Data.

| Fire Classification | Application of the Ball | Time to Activate (seconds) | Success/Failure |
|---|---|---|---|
| Class A | Wall Mount | 250.8 | Failure |
| Class B | Direct Throw | 3.6 | Failure |
| Class A | Wall Mount | 360.5 | Failure |
| Class B | Wall Mount | 321.3 | Failure |
| Class B | Wall Mount | 230.9 | Failure |
| Class A | Direct Throw | 4.1 | Failure |
| Class A | Direct Throw | 3.8 | Failure |
| Class B | Direct Throw | 4.3 | Failure |

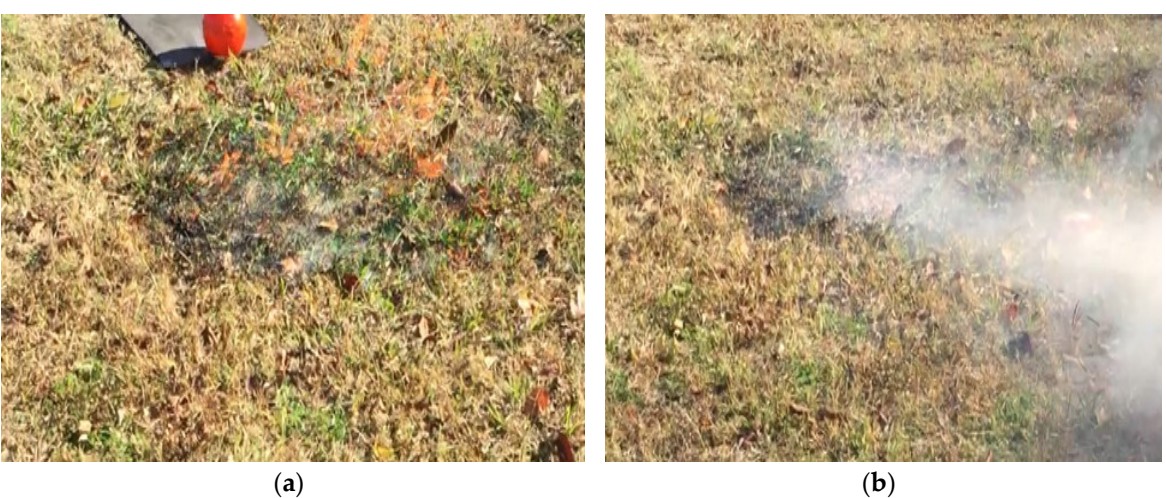

(**a**)　　　　　　　　　　　　　　　　　　　(**b**)

**Figure 4.** (**a**) Short grass fire before explosion; (**b**) short grass fire after explosion.

The fire was completely extinguished by means of the fire extinguishing ball. Considering the fact that the ball used for this experiment was the cheapest and smallest version of the fire extinguishing balls available at the marketplace, we believe that assisting firefighters during wildfires with fire extinguishing balls looks promising for the future. The 0.5 kg size ball extinguished a circle of 1-m diameter wildfire in our experiments, yet the 1.3 kg balls are claimed to extinguish a radius of 1.3 m.

After the experiments a total of eight questions were asked of two firefighters to obtain subject matter expert feedback. Both firefighters directly participated in the experiments. The questions included building fire and wildfire related questions. Their responses are illustrated next.

## 5. Subject Matter Experts' Feedback

Question 1: "Tell me what you think about the effectiveness of these balls with respect to extinguishing building fires."

In response to question 1, the firefighters strongly recommended use of a larger size fire-extinguishing ball and if possible to shoot multiple balls into a room in case of building fires. They thought that even this size of AFO ball might be effective on slowing down the building fire if thrown in batches. They also commented that if the balls were thrown from a higher elevation the powder might spread in a more circular fashion and might be more effective. This brought the research question of what the optimal elevation of dropping balls should be with the UASs. As it is dangerous to get the UASs close to fire because of the updrafts caused by the fire, it is important to investigate the optimum elevation as well as adjusting dropping strategies for weather (i.e., wind) that keeps the UASs safe, and the balls most effective. This research question is not addressed within the scope of this paper. Another comment from the firefighters was that the ball was obviously not effective against Class B fires.

Question 2: "How would you compare the after fire damage as a result of using these balls versus using water as fire suppressants?"

With respect to after fire damage of using the ball, firefighters thought the balls were very usable. One of them commented that: "...Water damage is devastative, compared to that, these balls won't cause any damage to the buildings…". They also commented about the fact that it is going to help save a lot of water. A rule of thumb to calculate the amount of water needed per minute to put out a fire is the area divided by three. Consider the fire demonstration cell, which was built to be 8 ft by 8 ft. That constitutes an area of 64 ft$^2$. Thus, extinguishing the 64 ft$^2$ cell requires more than 20 gallons per minute of water by the rule of thumb.

Question 3: "Tell me how these balls could be used for firefighting specifically for your city?"

The firefighters responded that the driver could throw these balls into rooms with a broken window while the rest of the crews were doing their own duties during a fire. Since the driver mostly sits idle during the fire, it would be an activity for the driver to help the fire extinguishing effort.

Question 4: "Tell me what you think of autonomous UASs shooting these balls into a building fire before you even arrive at the fire scene?"

The firefighters argued that breaking the windows to be able to throw these balls inside might cause the fire to get even worse due to the rapid intake of oxygen into the room. However, they recommended that it would be very beneficial to use them for those rooms with already broken windows. During a fire, a window can be broken by itself, or firefighters break them to change the direction of the fire. Thus, they thought it would be beneficial if UASs shoot these balls into a building through already broken windows or any openings before the first responders arrive at the fire scene. The main contribution would be slowing down the fire.

Question 5: "Tell me what you think of autonomous UASs bringing these balls to the fire scene and giving instructions through an audio system to people on how to use them before you even arrive at the fire scene?"

They did not favor this concept idea, since they thought it might cause liability issues. If a citizen gets injured while following the instructions the UAS is broadcasting, that would cause serious liability concerns. Especially thinking of how the people are under a panic situation while a fire is going on; there is a high risk involved in this concept.

Question 6: "Other than fire-extinguishing balls, tell me what materials or equipment would you like a UAS to bring to the fire scene during the fire and post-fire?"

The firefighters discussed that the most important equipment is thermal cameras. If UASs equipped with thermal sensors arrive at the fire scene before them and provide a live aerial thermal view of the fire; that would help them tremendously in terms of time to control the fire once they arrive on the scene. They also recommended UASs bring medical supplies and even food.

Question 7: "How would you like drones to broadcast the fire scene to you before you arrive at the fire scene via thermal cameras and regular video broadcasting?"

The firefighters partially responded to this question in the previous step as they discussed thermal cameras. Responding to Question 7, they also acknowledged the significance of regular video broadcasting before they arrive.

Question 8: "How could these balls be used during wildfires?"

The firefighters recommended the use of these balls for wildfire, specifically to prevent the fire spreading to buildings or towards significant fuel resources. They thought the ball was effective for

the short grass fire, and one firefighter said they would direct the UASs to certain points to drop these balls to mitigate the risk for buildings on the path of fire head.

As mentioned earlier, feedback from two firefighters does not provide any power to draw conclusions, but helped the research team change the main focus from building fires to wildfires. A further study might incorporate a mixed method including a survey and interviews by increasing the number of participating firefighters. This would help the research team to statistically be able to draw generalizations or conclusions with regards to the questions listed above.

Before the experiments, the main research objective was combatting high-rise building fires. Based on the findings of the experiments, the research team changed the key objective to mitigate the risk of wildfires spreading to buildings and potential fuel resources that could worsen the fire.

## 5. Discussion

Consider a swarm of 10 UASs, each carrying 10 of the 1.3 kg fire-extinguishing balls (each extinguishing a radius of 1.3 m). If they fly in a parallel fashion, and each of them drops the balls in a straight line one by one autonomously; that would constitute an area of 676 m$^2$ (=(1.3 * 2 * 10)$^2$) per sortie. If the UASs could be re-loaded with agile methods, it could be possible to prevent wildfire from spreading to buildings or critical directions, or to extinguish spot fires igniting during the main wildfire with this swarm size. If the swarm size increases and/or the number of balls per UAS increases, the effectiveness of the system can be increased exponentially. The research team is by no means claiming extinguishing a wildfire completely with the current state of the art of the UAS technology. Yet, our foremost objective is designing a system to assist firefighters during wildfires, especially those in wildland–urban interface zones. The conceptual system is as follows: (1) swarm of scouting UAS to detect spot fires and monitor the risk of wildfire approaching a building, fences, or firefighting crew via remote sensing, (2) communication UAS to establish and extend the communication channel between scouting UAS and fire-fighting UAS, and (3) a swarm of fire-fighting UAS autonomously traveling to the waypoints and dropping fire extinguishing balls. An algorithm will be developed and tested on simulation environment. The input from the visual and infrared sensor data from S-UAS, incorporated with wildfire models such as BehavePlus [6] and FlamMap [40], will be used to predict the prognosis of the fire behavior. The system will be tested in simulation environment for true alarm rates and timeliness for prediction of fire spread towards a building/fence, or firefighter. The research team has already completed the building of a scouting UAS and also a fire-fighting UAS with payload capacity of around 15 kg and time of flight of around 25 minutes. In addition, a mechanism attachable to the UAS that can carry fire-extinguishing balls has been completed. Figure 5 shows the scouting UAS, while its components are listed on Table 4. Table 5 shows the components of the fire-fighting UAS.

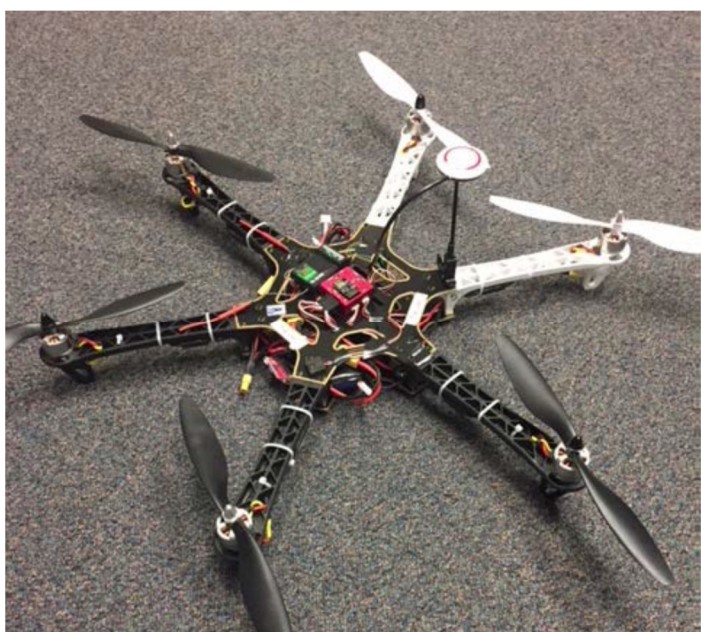

**Figure 5.** Scouting unmanned aircraft system (UAS).

**Table 4.** Components of UAS for remote-sensing operations.

| Component | Description |
| --- | --- |
| Frame | Composite Hexacopter frame 550 mm with integrated printed circuit board |
| Motors | 2212–920 KV Brushless |
| ESCs | 20 Amp 2–4S Lipo Compatible (OPTO) |
| Flight Controller | Pixracer |
| GPS | NEO-M8N & compass |
| Telemetry | 2.4 GHz Wi-Fi Module |
| RC Transmitter | FS-i6S 2.4 GHz 10 ch transmitter & 6 ch receiver |
| Battery | 2200 mAh 3S 20C Lipo |
| Propellers | 10 × 4.5 SF Nylon Composite |
| Raspberry Pie | Raspberry Pi Zero Wireless (1 GHz Single-core CPU, 512 MB RAM) |
| Thermal Sensor | SUNKEE DS18B20 Temperature Sensor |
| Camera | Arducam 5 Megapixels 1080p Sensor OV5647 Mini Camera |

**Table 5.** Components of fire-fighting UAS carrying fire-extinguishing balls.

| Component | Description |
| --- | --- |
| Frame | Tarot T18" 1270 mm octocopter foldable frame |
| Motors | Tarot 5008/340 KV MultiCopter Brushless TL96020 |
| ESCs | YEP 60A (2–6S) SBEC Brushless |
| Flight Controller | PixHack V3 |
| GPS | M8N |
| Telemetry | WiFi |
| RC Transmitter | Flysky FS-i6S 2.4 G 10 CH AFHDS 2A Transmitter With FS-iA10B Receiver |
| Battery | Multistar High Capacity 20,000 mAh 6S 10C Multi-Rotor Lipo Pack |
| Propellers | Tarot 1555 |
| Raspberry Pie | Raspberry Pi Zero Wireless (1 GHz Single-core CPU, 512 MB RAM) |

The battery can supply 200 amps (=20A * 10C) continuous current and 400 amps (=20A * 20C) peak current. Each motor can pull 25 amps (=200A/8) continuously. According to manufacturer's thrust tests, each motor provides a thrust of 3.01 kg with 1855 carbon propellers, at 22.2 V, 23.1 amps. Overall, eight motors will give a thrust of around 24 kg. Subtracting the weight of the frame, motors, ESCs, and other components of the UAS, there is a room for almost 15 kg to add the fire-extinguishing

balls and the ball-dropping mechanism. This means the firefighting UAS can carry up to 10 fire-extinguishing balls if needed. A ball-release mechanism was designed and built as a mechatronic system attachable to the firefighting UAS. The system consists of electronic components such as microcontroller, power supply, sender and receiver, and motor as well as a mechanism to carry and release the ball and its connections to the fire-fighting UAS. Figure 6 shows the built prototype including the ball-releasing mechanism and attached mechatronic components, as well as the remote starter button attached to a signal sender. This mechanism will be attached to the fire-fighting drone to perform the future experiments of the research in order to test the conceptual design with timber litter, short needle litter, and chapparal vegetation-based experimental settings. By taking the observations from the prototype into consideration, another mechanism that would carry at least three balls will be designed and built.

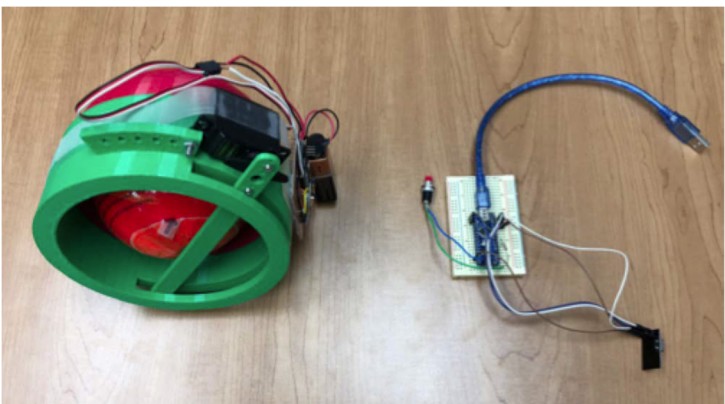

**Figure 6.** Prototype of a ball-release mechanism (**left**) and remote starter circuit (**right**).

## 5. Conclusions and Future Work

This article demonstrates the results of controlled experiments to test the efficiency of fire-extinguishing balls for firefighting. These experiments were conducted as part of an ongoing research project the goal of which is to design a system of UAVs incorporating remote sensing, and fire-extinguishing balls to control wild fires. The results of the experiments show that fire-extinguishing balls might be an effective and efficient tool to aid wildfire fighting if a swarm of drones could drop them to optimal points, in optimal numbers, on time. Remote-sensing technology will be needed to determine points of attack. A multi-institutional research is continuing in order to design a system where swarm of UASs detect a wildfire approaching buildings, fences, or firefighters, and dropping fire extinguishing balls. Future work involves developing the swarm platforms for the remote-sensing and fire-fighting components, upgrading the dropping mechanism, and developing wind trajectory models to experiment optimal dropping methods over timber litter, short needle litter, and chapparal vegetation models. There are some limitations in this study, mainly due to resource availability at the time of the experiments. The research team used the cheapest fire-extinguishing ball model in the experiments. These balls are only 0.5 kg, which means that they hold less than half of the mass of the power extinguishing powder compared to the 1.3 kg models. In further studies, 1.3 kg or custom made heavier balls could be tried for building fires. Moreover, only a fire in short grass was tested due to the time and budget availability of the City of Commerce, TX, fire department. As mentioned, timber litter, short needle litter, and chapparal vegetation models should be tested to reach any generalization for wildfires. Despite the limitations, the first key finding from this study is that 0.5 kg fire extinguisher balls are not effective for either Class A or B fires, but larger sized balls should be re-tested. Secondly, even the smallest size of the fire-extinguishing balls from the manufacturer who offers the lowest price worked effectively to extinguish a short grass fire that reached an area of one meter diameter. This is a promising result for further studies that will be conducted for developing the proposed drone-assisted wildfire-fighting system.

**Author Contributions:** Conceptualization, B.A., E.S., J.T. and M.S.; methodology, B.A., E.S., J.T. and M.S.; software, B.A.; validation, B.A., E.S., J.T. and M.S.; formal analysis, B.A., E.S., J.T. and M.S.; investigation, B.A.,

E.S., J.T., and M.S.; resources, B.A.; data curation, B.A.; writing—original draft preparation, B.A.; writing—review and editing, B.A., E.S., J.T. and M.S.; visualization, B.A., E.S., J.T. and M.S.; project administration, B.A.; funding acquisition, B.A., E.S., J.T. and M.S.

**Funding:** This research received no external funding.

**Acknowledgments:** The authors would like to thank Texas A&M Engineering Experiment Station for the support for the startup of the project. Authors also would like to thank the Kashmir World Foundation, and Texas A&M University-Commerce Electrical Engineering faculty Mr. Gerald Patrick Carter for their contribution to the drone building component of the project. In addition, the authors would like to thank Jacksonville University Engineering Department students Mr. Sergio A. Aponte and Mr. Diego J. Diaz Sanchez and faculty Dr. Maria Javaid for their contributions in building the prototype of the ball-releasing mechanism.

**Conflicts of Interest:** The authors declare no conflict of interest.

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
