# Peer review of "Use of Fire-Extinguishing Balls for a Conceptual System of Drone-Assisted Wildfire Fighting"

_drones, doi:10.3390/drones3010017_

Round 1
Reviewer 1 Report
The research would have added value if the concept of protecting fire crews was better explained. Every year way too many wildland firefighters die because of rapidly moving fire fronts
. Drones would be a more valuable assist if they could track firefighters and flame spread to notify those in immediate danger.
Is there research into this area of protecting firefighters?
Author Response
Response to Reviewer 1 Comment
Point 1: The research would have added value if the concept of protecting fire crews was better explained. Every year way too many wildland firefighters die because of rapidly moving fire fronts. Drones would be a more valuable assist if they could track firefighters and flame spread to notify those in immediate danger.
Is there research into this area of protecting firefighters?.
Response 1: This is a great point. The research team would like to thank you for sharing this amazing idea with us.
We have been looking at the risk of fire spread towards buildings and fences, but we will also incorporate risk of flame spread towards firefighters. The remote sensing system will be designed to also detect firefighters to notify those in immediate danger. This will be done as part of the deep learning algorithms in the remote sensing design. Even though it is out of the scope of this specific paper, it is added to the conceptual design and explained in the following lines: 22, 173-176, 216, 367, 373, 410
After further review of the literature, we could not detect any research out there that has considered this concept neither.
Once again, we would like to thank you so much for your valuable feedback. We strongly believe this change improved the quality of our work significantly.

Reviewer 2 Report
The topic of the article is very relevant, the use of drones in all areas of life is timely, so it is also worth looking at the possibilities in case of forest fires. Authors tried to combine three different topic, as well as the problem of forest fires, effectiveness of explosive extinguisher suppressing forest fire (ground fire) and applications of drone. Finding the optimal is not easy because each topic is very interesting and also published in different journals; even combination of them were already discussed in different papers even more as mentioned in the reference list.
1. There are some papers which are typically focuses on the UAV/UAS/drone applications supporting forest fire management; referring on some of theme are required. Pl. Conceptual Approach of Measuring the Professional and Economic Effectiveness of Drone Applications Supporting Forest fire Management (doi: 10.1016/j.proeng.2017.12.132)
2. There are companies which make researches for bombing water using drone (Nitrofirex, www.nitrofirex.com; Singular Aircraft: http://singularaircraft.com/ ) Referring to them is required to give more complex review about the topic and raise the quality.
3. There are companies producing explosive extinguisher typically for suppressing forest fire (pl. Matmodel Beaextin, dawhois.com/site/matmodel.com.html in Spain but there is another in France too). There are efforts to use explosive hose also. Referring to them is required to give more complex review about the topic and raise the quality.
4. Dropping fire balls using UAV/drone generating prescribed fire was already mentioned in some conferences, however in book: Jerry LeMieux: Introduction to Unmanned Systems, Chapter 11. Restas: Firefighting applications, pp: 243-244. ISBN-13: 978-1480150836. Referring to them is required to give more complex review about the topic and raise the quality.
5. A review papers from "Drone" or other journal is required; example: doi: 10.1016/j.paerosci.2017.04.003 . Referring to them is required to give more complex review about the topic and raise the quality.
6. I am afraid Figure 1 is not correct or not enough clear. It is the rate of fire spread v. fire intensity or rate of suppression speed v. fire intensity?
Author Response
Response to Reviewer 2’s Comments
As the research team, we would like to thank you for you valuable reviews and suggestions. We incorporated all your feedback to the paper. Below, please see line by line explanation of the changes.
Point 1: There are some papers which are typically focuses on the UAV/UAS/drone applications supporting forest fire management; referring on some of them are required. Pl. Conceptual Approach of Measuring the Professional and Economic Effectiveness of Drone Applications Supporting Forest fire Management (doi: 10.1016/j.proeng.2017.12.132)
Response: We added this reference. Please see line 126. We also included several additional studies on line 122 to 130. We also want to clarify that there are already several studies included that focused on use of UAS for forest fire fighting which were collected via an extensive literature research.
Point 2: There are companies which make researches for bombing water using drone (Nitrofirex, www.nitrofirex.com; Singular Aircraft: http://singularaircraft.com/ ) Referring to them is required to give more complex review about the topic and raise the quality.
Response: We added these companies and another company to the paper. Please see line 184-186.
Point 3: There are companies producing explosive extinguisher typically for suppressing forest fire (pl. Matmodel Beaextin, dawhois.com/site/matmodel.com.html in Spain but there is another in France too). There are efforts to use explosive hose also. Referring to them is required to give more complex review about the topic and raise the quality.
Response: Thank you for this suggestion. Yet, we are specifically interested in fire extinguisher balls for this research project. That is our niche. In further studies, we might try to also use other explosive agents. We would still like to include the companies you suggested however they don’t even have a company website. We cannot access any work that can be referenced in a scholarly format. The link you provided is not accessible. We couldn’t find any information about this company. Could you give us more feedback on this?
Point 4: Dropping fire balls using UAV/drone generating prescribed fire was already mentioned in some conferences, however in book: Jerry LeMieux: Introduction to Unmanned Systems, Chapter 11. Restas: Firefighting applications, pp: 243-244. ISBN-13: 978-1480150836. Referring to them is required to give more complex review about the topic and raise the quality.
Response: The reference to this book will be added to line 211, once the book is received. The estimated delivery date is Feb 7th.
Point 5: A review papers from "Drone" or other journal is required; example: doi: 10.1016/j.paerosci.2017.04.003 . Referring to them is required to give more complex review about the topic and raise the quality.
Response: We referenced the review paper you mentioned. Please see line 112 to 114, which is reference number 10. Also, we had a review paper which was our reference number 20.
Point 6: I am afraid Figure 1 is not correct or not enough clear. It is the rate of fire spread v. fire intensity or rate of suppression speed v. fire intensity?
Response : Thank you for the concern. We explained the figure and its association to our research more clearly in the introduction. The variables here are explained starting from line 55 until the end of introduction section.
The chart’s horizontal axis is heat per unit area, versus the vertical axis is rate of spread as labelled on the axis. This is a typical fire characteristics model. Please see our references 5 and 6. The key here is short grass fires represents the region where rate of spread is high but heat per unit area is low. So, when we experiment the short grass fire, we are actually experimenting the wildfires with high rate of spread (chain/hr) and low heat per unit area. This has already been accomplished in our research. The next steps which is mentioned later is looking at the other regions of this chart to make it more applicable for a wider spectrum of types of wildfires. We tried to better explain this in the introduction section.
Once again, we would like to thank you so much for all your valuable feedback. We strongly believe these changes improved the quality of our work.

Reviewer 3 Report
Summary
The manuscript presents the results of a series of experiments that have been conducted as part of a case study to evaluate the capabilities of fire extinguishing balls. From the observed results, the authors conclude that fire extinguishing balls have a potential to facilitate the fighting of wildfires (as opposed to apartment fires). The presented case study has been carried out as part of the design of a conceptual system to support a drone-assisted wildfire fighting, which is briefly sketched in the paper at hand.
Strengths and Weaknesses
The manuscript addresses a practically relevant topic that has not been examined sufficiently yet. Gaining deeper insights into the effectiveness of fire extinguishing balls might indeed have the potential to open up new opportunities to fight wildfires. I can also follow the authors’ argumentation that fire extinguishing balls might be particularly suitable for a shipment with drones as they are lighter and better portioned than, for instance, water (as the traditional fire suppressant).
Notwithstanding these positive aspects, the intended contribution is undermined by several issues and shortcomings, which need to be carefully addressed before a publication of the manuscript can be considered. I will summarize my main points of critique below:
Study design (case study vs. controlled experiments). To start with, I am not convinced that the authors indeed conducted a case study. According to Yin (Case study research: Design and methods, 3rd ed., Thousand Oaks, CA, Sage 2003), a characteristic trait of a case study is that a phenomenon is examined in its natural, real-life context. However, the authors of the manuscript did not examine the application of fire extinguishing balls in a real firefighting scenario. Instead, they conducted a series of experiments to approximate a real-life setting. As this is a different form of research approach, the paper IMHO needs to be reframed accordingly. When conducting (controlled) experiments, it particularly must be described how internal validity has been maximized and the real-world scenario has been approximated. In particular, the authors need to argue in how far igniting an (apparently) arbitrary piece of grassland to start a “short grass fire” (p. 7) is suitable to simulate a wildfire or if care has been taken to identify a representative piece of land. Otherwise, it remains unclear if the obtained results are valid, i.e. if they allow a generalization to wildfires. Moreover, they need to better describe the conducted task and the characteristics of the interviewed experts. There is abundant literature on how to conduct and describe experiments (and interviews). I would recommend structuring the presentation according to such guidelines.
Study focus (apartment fires vs. wildfires). The authors position the presented study as a part of a larger endeavor to design a system to fight wildfires. Nevertheless, a major part of the study (i.e., most of the experiments and the interview questions) is devoted to examining the applicability of fire extinguishing balls to fight apartment fires. Several shortcomings result from the current positioning of the study: First, a major part of the manuscript does not seem to be connected to its actual topic. Second, the part of the study that is devoted to examining wildfires is not nearly as detailed as the experiments focusing on apartment fires. As a consequence, the scientific contribution in this direction appears to be limited.
Study results (validity and originality). With respect to the topic of the manuscript, the authors have merely shown that fire extinguishing balls can be suitable to terminate a small-scale grass fire. It remains unclear if this observation can be transferred to other forms of vegetation or fires of a larger-scale with different temperatures and different formations of pockets of embers. Furthermore, only one out of eight interview questions seemed to address the relevant field of study (wildfires). I am not sure if the presented results merit a publication in a scientific journal. Bluntly put, I would have expected a test of various scenarios, in which the type of vegetation and the scale of the fire varies. Also, I would have expected a focused interview to gain a deeper insight into the potential of fire extinguishing balls for the extermination of wildfires. The authors should IMHO consider repositioning the manuscript (and the focus of the study) accordingly.
Journal relevance. As described above, the manuscript mostly focuses on evaluating the applicability of fire extinguishing balls. Although this endeavor is part of a development of a larger, drone-based system to fight wildfires, details about this conceptual system are hardly provided. The paper admittedly also contains a presentation of a mechanism to carry fire extinguishing balls using a drone. This mechanism does not appear to be the result of a scientific (but rather an engineering) process, however. It is also not systematically tested and compared to existing approaches that are discussed as related work in the manuscript. As a consequence, the scientific merit of the presented mechanism remains unclear largely unclear. I accordingly recommend strengthening the discussion of the conceptual system and presenting more details. IMHO, reframing the manuscript to strengthen the topical relationship to the targeted journal is an important housekeeping task that needs to be completed before a publication should be considered.
Verdict
All in all, I find that the manuscript addresses an interesting and practically relevant topic that might potentially fit well into the journal. The current manuscript needs to be reframed in various ways to shape the potential of the presented research, though.
Author Response
Response to Reviewer 3’s Comments
As the research team, we would like to thank you for you valuable reviews and suggestions. We incorporated your feedback to the paper. Below, please see line by line explanation of the changes.
Point 1: Study design (case study vs. controlled experiments). To start with, I am not convinced that the authors indeed conducted a case study. According to Yin (Case study research: Design and methods, 3rd ed., Thousand Oaks, CA, Sage 2003), a characteristic trait of a case study is that a phenomenon is examined in its natural, real-life context. However, the authors of the manuscript did not examine the application of fire extinguishing balls in a real firefighting scenario. Instead, they conducted a series of experiments to approximate a real-life setting. As this is a different form of research approach, the paper IMHO needs to be reframed accordingly. When conducting (controlled) experiments, it particularly must be described how internal validity has been maximized and the real-world scenario has been approximated. In particular, the authors need to argue in how far igniting an (apparently) arbitrary piece of grassland to start a “short grass fire” (p. 7) is suitable to simulate a wildfire or if care has been taken to identify a representative piece of land. Otherwise, it remains unclear if the obtained results are valid, i.e. if they allow a generalization to wildfires. Moreover, they need to better describe the conducted task and the characteristics of the interviewed experts. There is abundant literature on how to conduct and describe experiments (and interviews). I would recommend structuring the presentation according to such guidelines.
Response to Point 1: We agree with your comment with regards to conduction of case study versus controlled experiments. What we conducted is indeed ‘controlled experiments’ rather than a case study. The changes associated to this can be found on the lines: 2-5, 27-28, 222, 226, 227, 250, 266, 403.
Considering the internal validity of the wildfire experiments, it should be noted that we only looked at short grass fire for this part of the research. We believe we did not have a good job of explaining this clearly in the initial submission. Future experiments will include an independent variable category, which is type of vegetation. This factor will have not only short grass but also 3 other levels: short grass, timber litter, short needle litter, and chapparal. The reason behind selecting these 4 variables are explained in lines 81-87 and 250-255 clearly in this revised version of the manuscript. It is basically based on the surface fire behaviour model.
At this initial stage of our research, we could not design an experiment with statistical power for the wildfire experiments due to budget limitation and also the time and resource limitation of the fire department. That is why we did not include any hypothesis and claim to reject or not reject. We are just providing our controlled experiments with using short grass as the only experimental condition. In both trials, we used the same type of fire extinguishing balls, same method to throw the ball into the short grass fire, the area had the same external humidity and temperature (The trials were conducted right after each other with the same personnel in the same short grass land)
Another key here, which is mentioned in lines 276 to 280, is that the short grass fire was extinguished by the smallest and cheapest size of the fire extinguishing balls. The future experiments (out of the scope this paper) are going to include the size of the ball and the model as an independent variable as well.
We changed the term interview into subject matter expert feedback, as these were not formal interviews that you thought of. We don’t want to use the term interview. However, we believe that experiments were explained in a scientific manner.
Point 2: Study focus (apartment fires vs. wildfires). The authors position the presented study as a part of a larger endeavor to design a system to fight wildfires. Nevertheless, a major part of the study (i.e., most of the experiments and the interview questions) is devoted to examining the applicability of fire extinguishing balls to fight apartment fires. Several shortcomings result from the current positioning of the study: First, a major part of the manuscript does not seem to be connected to its actual topic. Second, the part of the study that is devoted to examining wildfires is not nearly as detailed as the experiments focusing on apartment fires. As a consequence, the scientific contribution in this direction appears to be limited.
Response to Point 2: We tried to explain that our initial key objective was to combat building fires with drones equipped with fire extinguishing balls. The detail on wildfire experiments were increased in this revised version. Please see lines 250-257
Point 3: Study results (validity and originality). With respect to the topic of the manuscript, the authors have merely shown that fire extinguishing balls can be suitable to terminate a small-scale grass fire. It remains unclear if this observation can be transferred to other forms of vegetation or fires of a larger-scale with different temperatures and different formations of pockets of embers. Furthermore, only one out of eight interview questions seemed to address the relevant field of study (wildfires). I am not sure if the presented results merit a publication in a scientific journal. Bluntly put, I would have expected a test of various scenarios, in which the type of vegetation and the scale of the fire varies. Also, I would have expected a focused interview to gain a deeper insight into the potential of fire extinguishing balls for the extermination of wildfires. The authors should IMHO consider repositioning the manuscript (and the focus of the study) accordingly.
Response to Point 3: Yes, as mentioned in earlier responses, different vegetation types (which means different rate of spread, and heat per unit area, see figure 1), different elevation of drone to drop the balls, different ball sizes and models, and several other variables will be investigated in the ongoing research. Actually, these experiments are under design right now. However, we believe that the current results are sufficient for meriting a publication. The reason is that considering the use of the smallest size and cheapest model of the fire extinguishing balls, we were able to extinguish the short grass fire in all trials. The results of this stage of the research are actually pretty important for the ongoing research. It helped us to understand that it was a better idea to utilize this concept for wildfires rather than building fires.
Point 4: Journal relevance. As described above, the manuscript mostly focuses on evaluating the applicability of fire extinguishing balls. Although this endeavor is part of a development of a larger, drone-based system to fight wildfires, details about this conceptual system are hardly provided. The paper admittedly also contains a presentation of a mechanism to carry fire extinguishing balls using a drone. This mechanism does not appear to be the result of a scientific (but rather an engineering) process, however. It is also not systematically tested and compared to existing approaches that are discussed as related work in the manuscript. As a consequence, the scientific merit of the presented mechanism remains unclear largely unclear. I accordingly recommend strengthening the discussion of the conceptual system and presenting more details. IMHO, reframing the manuscript to strengthen the topical relationship to the targeted journal is an important housekeeping task that needs to be completed before a publication should be considered.
Response to Point 4: This mechanism has been built by following an engineering design process. The related work discussed in the manuscript for this type of mechanism are only undergraduate design projects as you can see in the background section. We actually went through their design in the thesis papers. However, the research team see no need on directly comparing our mechanism to those mentioned in the background. Our design works for what we need and our objective is increasing the number of balls it can carry as mentioned in the paper. We have made a few changes to the lines 388-398 to better clarify.
Once again, we would like to thank you so much for all your valuable feedback. We strongly believe these changes improved the quality of our work.

Round 2
Reviewer 3 Report
Summary
The manuscript presents the results of a series of experiments that have been conducted to evaluate the capabilities of fire extinguishing balls. From the observed results, the authors conclude that fire extinguishing balls have a potential to facilitate the fighting of wildfires (as opposed to apartment fires). The presented experiments have been carried out as part of the design of a conceptual system to support a drone-assisted wildfire fighting, which is described in the paper at hand.
Strengths and Weaknesses
This manuscript is a revision of an earlier version that I had commented upon. I will therefore focus on the main points of critique uttered in my previous review:
1. Study design (case study vs. controlled experiments). In response to the critique, the authors have dropped the term "case study" in favor of the term "experiment". This change in terminology is very welcome as it solves some of the issues mentioned in my earlier review (though it should also be implemented in the caption of Figure 3). IMHO, however, the authors still do not provide sufficient detail on how the described experiment setting was designed to approximate a real-life setting. This is especially the case for the wildfire scenario that turned out to be most important for the manuscript. This should be addressed before a publication of the manuscript is considered.
2. Study focus (apartment fires vs. wildfires). The authors position their study as a part of a larger endeavor to design a system to fight wildfires. Although wildfires are discussed in more depth now, a major part of the manuscript still is devoted to examining the applicability of fire extinguishing balls to fight apartment fires. In the revised version of the manuscript, the authors provide better arguments why apartment fires were also examined. IMHO, however, it still appears necessary to shift the focus more towards the fighting of wild fires, which are claimed to be the central topic of the paper.
3. Study results (validity and originality). In their response to the reviewers, the authors argue why the presented results are preliminary with respect to the fighting of wildfires. I agree and recommend that the manuscript should be augmented with a thorough discussion of the limitations of the study. Also, it should be discussed in how far the obtained results can be generalized to the various types of wildfires that are discussed in the manuscript. If the results cannot be generalized or transferred at all, the intended contribution IMHO remains questionable.
4. Journal relevance. In the revised manuscript, the authors provide more details about the development of a larger, drone-based system to fight wildfires. I acknowledge that the design of the system is still in an infant stage. Therefore, if the editors find the provided details to be sufficient, I will not object.
Verdict
All in all, I think that the authors have made progress to better position the conducted study. IMHO, some issues still need to be addressed before a publication should be considered.
Author Response
Second Round Response to Reviewer 3’s Comments
As the research team, we would like to thank you for your further valuable reviews and suggestions. We tried to incorporate your feedback to the paper. Below, please see line by line explanation of the changes.
Point 1: 1. Study design (case study vs. controlled experiments). In response to the critique, the authors have dropped the term "case study" in favor of the term "experiment". This change in terminology is very welcome as it solves some of the issues mentioned in my earlier review (though it should also be implemented in the caption of Figure 3). IMHO, however, the authors still do not provide sufficient detail on how the described experiment setting was designed to approximate a real-life setting. This is especially the case for the wildfire scenario that turned out to be most important for the manuscript. This should be addressed before a publication of the manuscript is considered.
Response to Point 1:
Figure 3 title has been fixed. Please see line:_238__
The wildfire experiments were explained in lines 239 until 246. We tried to explain with more detail. Please see lines: 245-251.
We understand your concern, but the wildfire experiments were only for short grass fires within the scope of this paper. That is why there is limited detail we can add. However, this is the type of short grass that you will see for majority of the houses in rural and sub-urban areas. This type of grass is very significant in our research, because it is the main connection of wildfires and houses. It is dry short grass as can be seen in Figure 4.
Point 2: Study focus (apartment fires vs. wildfires). The authors position their study as a part of a larger endeavor to design a system to fight wildfires. Although wildfires are discussed in more depth now, a major part of the manuscript still is devoted to examining the applicability of fire extinguishing balls to fight apartment fires. In the revised version of the manuscript, the authors provide better arguments why apartment fires were also examined. IMHO, however, it still appears necessary to shift the focus more towards the fighting of wild fires, which are claimed to be the central topic of the paper.
Response to Point 2: Because, these fire extinguisher balls are claimed to be effective in A,B,C classification of fires by their manufacturers, we wanted to examine if their claims were accurate. On the other hand, the manufacturers had no claims on the effectiveness of these balls with wildfires. Therefore, it was important for us to test for both building and wildfires. Please see lines 100-103.
What we are emphasizing is that the cheapest fire extinguishing ball model (and it’s the smallest size of the model) was used in our wildfire experiments, and it still completely extinguished the short grass fire in each trial. Firefighters were very excited and interested about this. Yes, we agree with you that focus could be shifted more towards wildfire fighting, but in this paper both aspect is important. There are cheap models out there, which we found that they didn’t work for Class A, B, or C fires in buildings. This is an important finding not just for academic research, but also for public safety. Using these cheap balls in houses or business on wall mounts, relying that they will save the house or business from fires doesn’t seem like a good idea after we saw the results of the experiments and discussion with firefighters. The fire extinguisher balls is an emerging technology and this paper makes the public aware that the AFO small size models might not be reliable. This paper clearly shows that. We used the AFO brand model, so our conclusions are for that model. But, what we found for wildfires is a different story. These balls worked well. We will still need to test them for other types of wildfires as we explained in future study.
Since our results considering building fires, and wildfires are both very important, the paper explains both building and wildfire experiments.
Point 3: Study results (validity and originality). In their response to the reviewers, the authors argue why the presented results are preliminary with respect to the fighting of wildfires. I agree and recommend that the manuscript should be augmented with a thorough discussion of the limitations of the study. Also, it should be discussed in how far the obtained results can be generalized to the various types of wildfires that are discussed in the manuscript. If the results cannot be generalized or transferred at all, the intended contribution IMHO remains questionable.
Response to Point 3:
Limitations are added and the key result of our experiments are explained with respect to generalization of results. Please see lines 410-422. Also line 25-27.
Point 4: Journal relevance. In the revised manuscript, the authors provide more details about the development of a larger, drone-based system to fight wildfires. I acknowledge that the design of the system is still in an infant stage. Therefore, if the editors find the provided details to be sufficient, I will not object.
Response to Point 4:
Thank you for your comment. We believe that the detail given is sufficient within the scope of this particular manuscript, as we will submit another manuscript with the results of our main detailed wildfire experiments and system design next year.
Once again, we would like to thank you so much for all your valuable feedback. We strongly believe these changes improved the quality and clarity of our work.
